# Control of Mechanical Properties of FRP (Fiber-Reinforced Plastic) via Arrangement of High-Strength/High-Ductility Fiber in a Blended Fabric

**DOI:** 10.3390/ma16145001

**Published:** 2023-07-14

**Authors:** Ji Hyun Kim, Bhum Keun Song, Joon Hyuk Song, Kyoung Jae Min

**Affiliations:** Convergence Research Division, Korea Carbon Industry Promotion Agency, 110-11 Banryong-ro, Jeonju 54853, Republic of Korea; jhkim@kcarbon.or.kr (J.H.K.); songjh@kcarbon.or.kr (J.H.S.); bluemy@kcarbon.or.kr (K.J.M.)

**Keywords:** CFRP, blended fabrics, FRP mixing, composite materials, mechanical properties

## Abstract

Carbon fiber-reinforced plastic (CFRP) has been widely investigated as a reinforcement material to address the corrosion and durability issues of reinforced concrete (RC). To improve the strain of FRP grids, we investigated the effect of single-fiber types, hybrid ratios, and stacking patterns on the strain of the composite materials. Blended fabrics in which different fibers are woven were used to further improve the strain of carbon fibers (CFs). In the blended fabrics, CFs with high tensile strength were mixed with high-strain glass fibers (GFs) or aramid fibers (AFs). Fibers with different mechanical properties were mixed to improve the strain without reducing the tensile strength of the composite materials. The fiber arrangement direction was controlled by CF/GF blended fabric. CFs are arranged in the direction parallel to the tensile load direction with no strength degradation, and GFs are arranged in the direction perpendicular to the increase in strain. Compared to the mechanical properties of the single CF composites, the fabrics obtained via an FRP mixing method proposed in this study showed an increase in the tensile strength by 7% from 568.17 to 608.34 MPa with no strength degradation and an increase in strain by 34% from 0.97% to 1.30%.

## 1. Introduction

Reinforced concrete (RC) is a composite material composed of steel rebars and concrete wherein steel rebars are used to reinforce the weak tensile strength of concrete. The durability of RC structures is known to rapidly degrade due to corrosion. The degradation of RC structures leads to an increased risk of accidents and increases the maintenance costs. Hence, to address the degradation of RC structures, efforts have been made to replace steel rebars that are vulnerable to corrosion with fiber composites that are highly resistant to corrosion [1,2,3,4,5,6,7].

Since the 1970s, studies have investigated the application of glass fiber-reinforced plastic (GFRP) as a reinforcement material for concrete. The interfacial adhesion between GFRP and concrete is high owing to the similarity in the strain values of GFRP and concrete. In addition, the potential of GFRP as a substitute for steel rebars has been confirmed [8]. However, the strength of GRFP is lower than that of steel rebars. When comparing the tensile strength of GFRP and steel rebar per the same area, the density of the steel rebar has a value about three times higher than the GFRP, and the tensile strength of the steel rebar has a relatively large value. On the other hand, when comparing the tensile strength of GFRP and steel rebar per the same weight, the tensile strength of the GFRP has a relatively large value. Moreover, by replacing GFRP with carbon fiber-reinforced plastic (CFRP), a higher tensile strength can be obtained.

Subsequently, CFRP grids with high strength were developed and they are used as reinforcement for concrete structures such as bridges and wall panels. The use of CFRP has been shown to increase the strength of RC structures [9,10]. However, the low strain value of CFRP leads to separation at the interface with concrete, resulting in the brittle failure of concrete. Therefore, it is necessary to increase the strain of composite materials by mixing high-strength carbon fibers with highly ductile aramid and glass fibers [11,12].

For example, Sun et al. [13] controlled the mechanical properties by mixing two types of single-fiber fabrics and changing the stacking pattern under a fixed fiber volume ratio of 50:50 *v*/*v*. The reason why the fiber mixing ratio was fixed at 50:50 *v*/*v* in the stacking pattern was that the mechanical properties of the fiber mixing were obtained close to the CF when the CF content was mixed at 60 wt.% or more. When the stacking pattern was changed from sandwich-stacking to cross-stacking (i.e., as the degree of mixing increased), the tensile strength increased by 23.42% from 354.39 to 437.15 MPa and the strain increased by 17.76% from 1.07% to 1.26%.

In recent years, in order to apply FRP reinforcement to concrete, the evaluations of FRP’s flame retardancy, acid/base safety, and low temperature properties have been actively studied [14].

Also, research is being conducted to develop FRP-based structures to improve buckling characteristics. Buckling properties of the FRP structure were carried out according to the size, shape, position, and number of holes of the open hole and the addition of the carbon nanotube (CNT) [15,16,17,18,19].

Herein, aiming to improve the strain of fiber-reinforced plastics (FRPs), we investigated the effects of fiber types, mixing ratios, and stacking patterns using single-fiber fabrics. Blended fabrics, in which different fibers are woven, were proposed to further improve the strain of carbon fibers. High-tensile-strength carbon fibers were blended with high-strain glass fibers or aramid fibers. Using a combination of fibers with different mechanical properties, we achieved an increase in the strain of the composite materials without a decrease in their tensile strength.

## 2. Materials and Methods

### 2.1. Materials

The carbon fiber (CF) T-300 was obtained from Toray (Tokyo, Japan). The CF had a tensile strength of 3.53 GPa, an elasticity of 230 GPa, a strain of 1.5%, and a density of 1.78 g/cm^3^. The CF fabric 3K (EK-3101, planar density: 200 g/m^2^) was obtained from Entra Korea (Busan, Republic of Korea). 

The glass fiber (GF) was obtained from CPIC; it had a tensile strength of 2.20 GPa, an elasticity of 70 GPa, a strain of 2.20%, and a density of 2.54 g/cm^3^. The GF fabric with a linear density of 300 Tex (TI-1313) was obtained from TEi Composites (Chang Hua, Taiwan). The aramid fiber (AF) HF200 with a linear density of 1500 denier was obtained from Kolon (Seoul, Republic of Korea); it had a tensile strength of 2.99 GPa, an elasticity of 94 GPa, a strain of 3.30%, and a density of 1.70 g/cm^3^. The AF fabric (TI-9013, planar density: 170 g/m^2^) was obtained from TEi Composites.

All figures and tables are cited in the main text as Figure 1, Table 1, etc. The reference fabrics CF8, GF8, and AF8 were obtained by stacking eight layers of CF, GF, and AF, respectively.

The hybrid fabrics were obtained by mixing two types of single-fiber fabrics. The fiber mixing ratio was fixed at 50:50 *v*/*v*. The alternating stacking pattern (CF/GF)4 and the sandwich-stacking patterns CF2/GF4/CF2 and GF2/CF4/GF2 were obtained using CF and GF fabrics. The alternating stacking pattern (CF/AF)4 and the sandwich-stacking patterns CF2/AF4/CF2 and AF2/CF4/AF2 were obtained using CF and AF fabrics.

The blended fabrics were obtained by weaving two types of fabrics and stacking eight layers of the resulting fabric. The blended fabric in which CFs and GFs were woven is referred to as CFGF8. CFGF8① denotes the fiber wherein the CFs are arranged in the direction parallel to the tensile load direction, and CFGF8② denotes the fiber wherein the CFs are arranged in the direction perpendicular to the tensile load direction.

### 2.2. FRP Manufacturing Method

FRPs were obtained via the resin infusion molding method. In this method, the materials, such as fabrics and laminated films, are stacked on the mold and resin is infused under vacuum conditions. Figure 2 shows the order of the materials stacked on the mold. The peel ply, fabric, peel fly, release film, mesh, resin/vacuum injection line, and vacuum film were stacked in sequence. The peel ply was used to ensure an easy separation of the composite material and mold after hardening. The release film had perforations at regular intervals and was used to ensure uniform infusion of the resin into the fabric. The mesh was used to improve the fluidity of the resin, thereby ensuring an even distribution over the materials. The vacuum film was double-layered so that the reduced pressure state could be maintained in the secondary vacuum film even after the primary vacuum film was infused with the resin. An epoxy resin (Kinetix, Molendinar, Australia, R-118, viscosity: 300 cp) was injected into the mold under a pressure of 0.8 bar. The resin and a curing agent (Kinetix, H-141) were mixed at a ratio of 1000:250 g. The ratio of resin and curing agent used in this paper was determined through preliminary experiments as an optimal ratio condition. The mold was subjected to heat treatment at 60 °C for 4 h.

### 2.3. Analysis Method

The fiber and resin weights of the FRPs were measured via thermogravimetric analysis (TGA, DSC 204, Dong-il Shimadzu, Seoul, Republic of Korea) in Figure 3. The heating rate was 5 °C/min in an air atmosphere and the heat treatment temperature was varied from 30 to 1000 °C. The resin and fiber underwent pyrolysis due to temperature changes, and their weights were measured. The thermal expansion coefficient of the FRPs was measured using a specimen size of 5 × 5 × 5 mm, a heating rate of 5 °C/min, a load of 0.05 N, and a temperature of −140–20 °C in accordance with ASTM E831.

The mechanical properties of the FRPs were evaluated using a universal testing machine (UTM, Instron 5982, Norwood, MA, USA). The tensile strength was measured using a specimen of size 25 × 250 mm and a test speed of 2 mm/min in accordance with ASTM D3039 [20]. The tensile modulus was measured in chord 0.1–0.3% attached with a strain gauge (square shape, size: 13 mm). The compressive strength was measured using a specimen measuring 12 × 140 mm and a test speed of 1.3 mm/min in accordance with ASTM D6641 [21] and attached with a strain gauge (square shape, size: 13 mm). The flexural strength was measured using a specimen of size 12.7 × 127 mm in accordance with ASTM D790-17 [22] and attached with a strain gauge (square shape, size: 13 mm). All mechanical property evaluations were repeated five times.

The fracture shapes of the FRPs according to the fiber type were analyzed at a low magnification of 30× using a field emission-scanning electron microscope (FE-SEM, SU8230, Hitachi, Tokyo, Japan). 

## 3. Results and Discussion

### 3.1. FRP Fiber/Resin Content Evaluation

Figure 4 shows the pyrolysis graphs of the FRPs. The fiber and resin contents were evaluated by measuring the changes in mass due to the pyrolysis of the FRPs. The pyrolysis temperatures of the matrix were 219.39 and 444.45 °C. The pyrolysis temperatures of the CFs and AFs were 584.51 and 464.23 °C, respectively. No mass reduction was observed in the GFs because GFs do not undergo pyrolysis at temperatures below 1000 °C. Measuring the AF content via pyrolysis was difficult because the pyrolysis temperatures of the resin and AFs were similar. Therefore, the AF content was evaluated using the fiber mass that was measured during the manufacturing of the FRPs. Table 2 shows the fiber and resin contents of the FRPs obtained from mass measurements and TGA. For CF8, GF8, and AF8, the differences in the values of the fiber and resin contents obtained from mass measurements and TGA were negligible. The ratios of the fiber volume to resin volume were largely similar (50:50). However, the volume and mass ratios differed depending on the fiber type owing to the difference in fiber density. The density of GFs is 2.54 g/cm^3^, which is approximately 1.4 times higher than that of CFs and AFs. Therefore, the fiber to resin volume ratio of GF8 was 50:50, but the fiber to resin mass ratio was approximately 70:30. The thermal expansion coefficient values of the developed composites are shown in Table 2. It was confirmed that the difference in the thermal expansion coefficient obtained by a single carbon, glass, and aramid fiber and its mixing fibers was not significant. Accordingly, the thermal expansion coefficient according to fiber mixing will not have a significant effect on thermal deformation and stress.

### 3.2. Tensile/Compressive/Flexural Properties of the Reference Fabrics

A comparison of the mechanical properties of the single-fiber composites with those of the hybrid and blended fabrics is shown in Table 3. CF8 exhibited the highest tensile strength owing to the high strength/high elasticity of the CFs. The tensile strengths of GF8 and AF8 were, respectively, 56 and 33% lower than that of CF8, but their strains were 35 and 66% higher. CF8 showed the highest compressive strength. The compressive strength of GF8 was 119% higher than that of AF8. This is because GFs have Si-O ion bonds that result in strong compression properties. In terms of flexural strength, AF8 exhibited the highest strain as the FRPs deformed significantly under flexural loading. The strain of AF8 was 162% higher than that of CF8 even though its flexural strength was 51% lower. CFRPs have been used as grids to provide tensile reinforcement for concrete. Hence, the mechanical properties of the hybrid and blended-stacking structures were evaluated and compared with those of CF8.

### 3.3. Tensile/Compressive/Flexural Properties of the CF-GF Hybrid/Blended Fabrics

Table 4 shows the mechanical properties of the fabrics obtained by a combination of CFs and GFs. The results of the tensile strength measurement indicate that the tensile strengths and strains of the blended fibers differed depending on the direction of CFs (parallel/perpendicular directions with respect to the tensile load axis direction). CFs are arranged in the direction parallel to the tensile load direction in CFGF8① and in the direction perpendicular to the tensile load direction in CFGF8②. The tensile strength of CFGF8① (608.34 MPa) was approximately 7% higher than that of CF8 (568.17 MPa), and its strain increased by approximately 34% from 0.97% to 1.30%. The tensile strength of CFGF8② (428.31 MPa) was approximately 25% lower than that of CF8 (568.17 MPa). Its strain, however, increased by approximately 76% from 0.97 to 1.71% owing to the high strain of GFs.

The tensile strengths of the hybrid fabrics were lower than those of CF8. The tensile strengths were obtained using the values based on the rule of mixtures (CF and GF volume ratio (*v*/*v*) 50:50, tensile strength: 409.87 MPa). The compressive strengths of CF8 and GF8 were 381.04 and 224.52 MPa, respectively. The compressive strengths of CFGF8① and CFGF8② differed owing to the differences in the fiber directions. The compressive strengths of CFGF8① and CFGF8② were 356.60 and 281.88 MPa, respectively. For the hybrid fabrics, there was no significant difference in tensile strength (mean: 388.13 MPa, deviation: 20 MP, within 5%) and compressive strength (mean: 296.42 MPa, deviation: 20 MP, within 6.82%) depending on the stacking pattern.

The FRPs deformed considerably under flexural loading and a significant difference in strain was observed depending on the addition of GFs. 

The strain values of the CF-GF blended and hybrid fabrics were higher than those of CF8. Moreover, CFGF8②, in which the GFs are arranged in the direction parallel to the flexural load direction, exhibited the highest strain value (3.99%). The tensile strength–strain and flexural strength–strain graphs shown in Figure 5 affirm these results. Unlike that in the hybrid fabrics, the increase in the strain value of CFGF8① was not accompanied by a decrease in the tensile strength. This can be attributed to the arrangement of CFs in the direction parallel to the tensile load direction, which affected the tensile strength; moreover, the hybrid effect—calculated as the ratio of the increased strain value due to the use of mixed fibers to the strain value of carbon fibers [23]—was significant due to the mixing of fibers with similar strains.

### 3.4. Tensile/Compressive/Flexural Properties of the CF-AF Hybrid/Blended Fabrics

Table 5 shows the mechanical properties of the fabrics obtained by a combination of CFs and AFs. The tensile strength of CFAF8① (516.30 MPa), which had the highest degree of CF-AF mixing, was approximately 9% lower than that of CF8 (568.17 MPa). The compressive strengths of CF8 and AF8 were 381.04 MPa and 102.72 MPa, respectively. Similar to the behavior observed in the blended CF-GF fabrics, the compressive strengths of CFAF8① and CFAF8② differed considerably. The compressive strength of CFAF8①, in which the CFs are arranged in the direction parallel to the tensile load direction, was 300.95 MPa; the compressive strength of CFAF8②, in which the CFs are arranged in the direction perpendicular to the tensile load direction, was 146.35 MPa. The flexural strain of CFAF8② increased by 174.35% from 1.91 to 5.24%.

The mechanical properties of the fibers stacked on edges significantly affect the results of flexural strength measurements; hence, a significant difference was observed in the strain depending on the stacking pattern. The strain value of AF2/CF4/AF2 in which the AFs were stacked on the edges was approximately 82.20% higher.

The tensile strength–strain and flexural strength–strain graphs shown in Figure 6 affirm these results. When the degree of mixing increased, the tensile strength increased while the strain decreased because the properties of the CFs were dominant. Conversely, when the degree of mixing decreased, the tensile strength decreased while the strain increased significantly because the properties of the AFs were dominant. 

In the CF-GF mixed stacking, the mean tensile strength was 440.21 MPa, with a deviation of 97 MPa (within 22%), and the mean tensile strain was 1.22%, with a deviation of 0.30% (within 25.23%). In the CF-AF mixed stacking, the mean tensile strength was 426.40 MPa, with a deviation of 52 MP (within 12.13%), and the mean tensile strain was 1.08%, with a deviation of 0.32% (within 29.50%).

In the CF-GF stacking, the average tensile strength increased significantly and the variation in strain due to the variation in the stacking pattern was minimal. In the CF-AF stacking, however, the average tensile strength did not increase significantly but the strain values differed considerably depending on the stacking pattern. The large difference in the strain depending on the stacking pattern can be attributed to the large difference in the strain values of the fibers. Therefore, mixing two fibers with similar strain values led to a positive hybrid effect, thereby improving the strain. This indicates that mixing CFs and GFs, which have similar strain values, is more effective in improving mechanical properties.

As a result of confirming the results of previous studies, Sun et al. [13] found that the tensile strength calculation based on the rule of mixture of the hybrid fabric of CF ([C]_8_) and BF ([B]_8_) has 459.115 MPa at the mixing volume ratio of 50:50 *v*/*v* and the tensile strengths of CF and basalt fiber (BF) are 504.73 MPa and 413.50 MPa, respectively. The tensile strength obtained by the hybrid stacking patterns of CF and BF is shown in the following Table 6, and these values are obtained as an approximation of the rule of mixture.

In addition, Margabandu et al. [24] found that the tensile strength calculation based on the rule of mixture of the hybrid fabric of CF ([CC]_s_) and Jute cottage([JJ]_s_) has 178.075 MPa at a mixing volume ratio of 50:50 *v*/*v* and the tensile strength of CC and JJ of 301.17 MPa and 54.98 MPa, respectively. The tensile strength obtained by hybrid stacking patterns of CF and JJ was obtained as an approximation of the rule of mixture, and does not exceed the tensile strength of CF.

On the other hand, Chen et al. [25] obtained the FRP using CF (CC) and GF (GG) blended fabric prepreg. C90G0 was positioned with CF in the longitudinal direction and GF in the latitude direction, the tensile strength of C90G0 was obtained similarly to the tensile strength of CF, and the tensile modulus was further improved. As a result of confirming the reason for the increase in the mechanical properties by simulation, it was reported that the tensile load was uniformly distributed at the intersection of CF and GF, and the sustainable strength of the tensile load was increased. In the above paper, a composite material was developed using prepreg uniformly impregnated with resin, and the ratio of fiber and resin was about 60:40 *v*/*v*. In comparison, in this study, a composite was developed by the infusion method, and the ratio of fiber and resin was about 50:50 *v*/*v*. The tensile strength of the composite developed by the prepreg molding method had a relatively high fiber content compared to the infusion method, and the tensile strength was shown to be a high value.

### 3.5. Evaluation of the FRP Fracture Shapes

Figure 7 shows the FRP fracture surfaces of the different fibers after tensile load measurements. The strain values of CF8, GF8, and AF8 were 0.97%, 1.31%, and 1.61%, respectively. In the composite containing CFs (which have low ductility), fracture occurred as if the CFs were cut. In composites containing GFs or AFs (which have high ductility), the fluff of the fibers was observed because the fibers were cut after they were elongated to the maximum strain threshold. AFs, which have a higher strain value than GFs, showed more fluff. In composites containing CFs and GFs, the difference between the strains of CFs and GFs was approximately 35%, and no significant difference in the fiber cut length was observed on the fracture surface. In composites containing CFs and AFs, however, the difference between the strains of CFs and AFs was approximately 66%, and the fluff due to the longer fiber cut length of AFs covered the fracture surface of the CF. This indicates that the difference in the breaking points of the fibers with different strain values affects the increase in the tensile strength and strain of composites.

### 3.6. Difference in Tensile Strength Depending on the Fiber Direction

The results of the aforementioned analysis indicate that both tensile strength and strain increased when CF-GF mixing was used, in contrast to the case when CFs alone were used. Accordingly, we attempted to determine the cause of this increase in tensile strength and strain. Table 7 shows the change in the length of grid composites in the parallel/perpendicular directions with respect to the tensile load direction along with the vertical difference (є_y_) measured from the Gauge length vertical length change.

In CF-GF blended-stacking, the force acting on the fibers is divided along the directions parallel and perpendicular to the tensile load direction owing to the arrangement of the fibers. When the high-strength CFs are arranged in the direction parallel to the tensile load direction and the GFs are arranged in the perpendicular direction, the CFs directly affect tensile strength and the GFs affect the deformation in the perpendicular direction. When GFs with high strain were arranged in the direction perpendicular to tensile load direction, the vertical difference (є_y_) increased from 0.12 to 0.19.

No decrease in the tensile strength was observed when CFs were arranged in the direction parallel to the tensile load axis, and the strain was increased due to the arrangement of high-strain GFs in the direction perpendicular to the tensile load direction.

Thus, both tensile strength and strain can be improved by utilizing the anisotropic characteristics of the composite. Hence the fabric with high-strength CFs in the direction parallel to the tensile load and high-strain GFs in the direction perpendicular to the tensile load exhibit increased tensile strength and strain compared to CF8.

## 4. Conclusions

In this study we investigated the effects of single-fiber types, hybrid ratios, and stacking patterns on the strain of FRP grids. Blended fabrics in which different fibers are woven together were used to improve the strain of CFRP grids. In the blended fabrics, high-tensile strength CFs were mixed with high-strain GFs or AFs.

(i) In CF-GF stacking, the average tensile strength increased significantly and the difference in strain was minimal depending on the stacking pattern. In CF-AF stacking, in contrast, the average tensile strength did not increase significantly, but the difference in strain was large dependent on the stacking pattern. The large stacking pattern-dependent difference in strain was attributed to the large differences between the strain values of the fibers. In the mixture of CF/GF and CF/AF, there was a difference in the improvement of mechanical properties depending on the fiber type. The mechanical properties of CF/GF with a strain difference of less than 50% between fibers have been further improved.

Hence, when two fibers with similar strain values were mixed, they had a positive impact on the hybrid effect and improved the strain. Therefore, mixing CFs and GFs, which have similar strain values, is more effective in improving mechanical properties.

(ii) The vertical difference (є_y_) increased from 0.12 to 0.19 when the high-strength CFs were arranged in the direction parallel to the tensile load direction and the high-strain GFs were arranged in the direction perpendicular to the tensile load direction. Accordingly, the tensile strength increased from 568.17 to 608.34 MPa and the strain increased by approximately 34% from 0.97 to 1.30%.

The tensile strain could be increased without a degradation in the tensile strength by changing the mixing pattern of the high-strength CFs and high-ductility GFs. Composite materials with such properties are expected to be used in FRP grids that require materials with improved strength and strain. Also, the composites developed in this study can be applied to the form of columns and pipes. It is expected that it will be easy to increase tensile strength by arranging CF in the direction parallel to the tensile or compressive load direction, and to use it in the part where bending and deformation are required in the shape of the composite material by arranging GF in the perpendicular direction.

## Figures and Tables

**Figure 1 materials-16-05001-f001:**
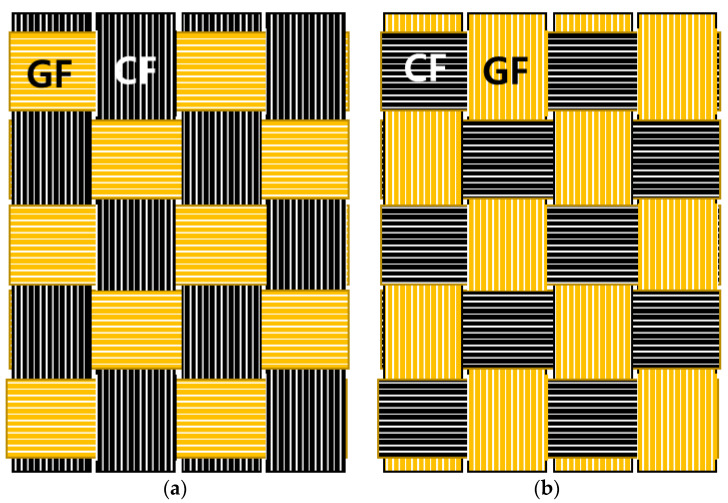
Fiber directions in the blended CF-GF fabrics. (**a**) CFGF①, (**b**) CFGF②.

**Figure 2 materials-16-05001-f002:**
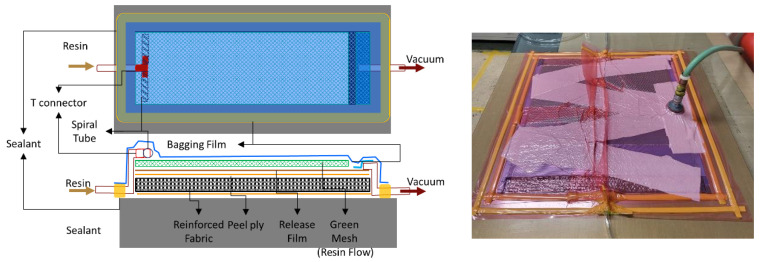
Stacking sequence used during the resin infusion molding method and the mold geometry.

**Figure 3 materials-16-05001-f003:**
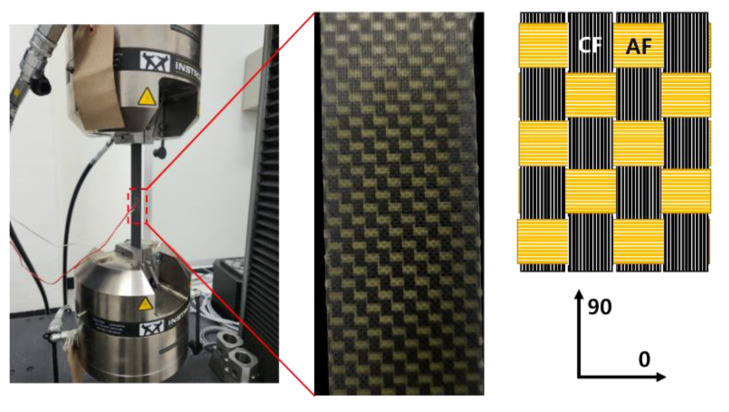
Tensile test setup for blended composite product.

**Figure 4 materials-16-05001-f004:**
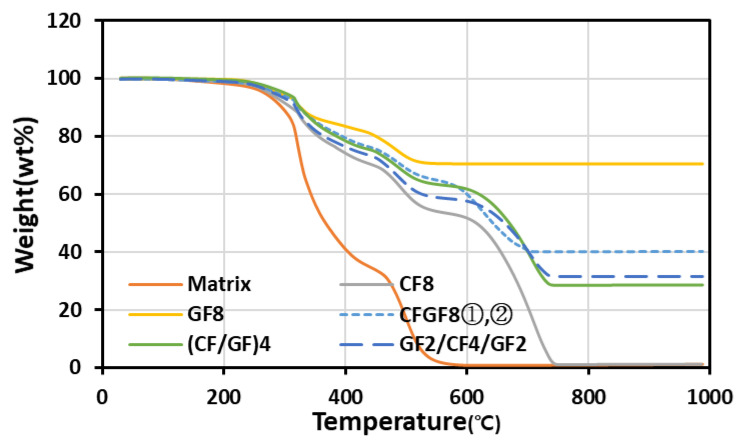
Results of the TGA evaluation of the FRPs.

**Figure 5 materials-16-05001-f005:**
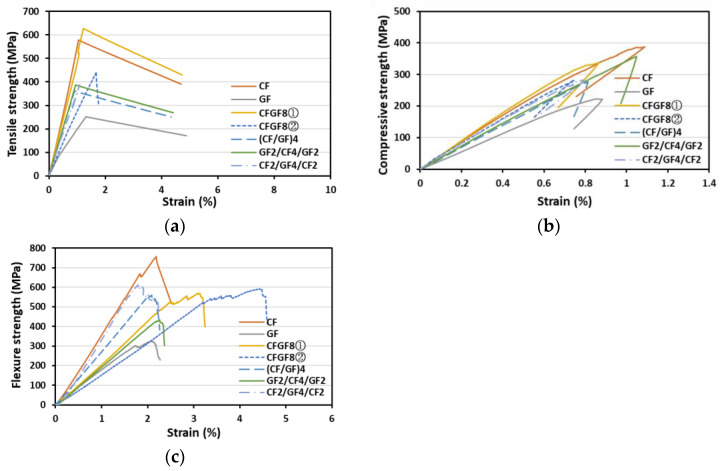
(**a**) T-S, (**b**) C-S, and (**c**) F-S graphs for the CF-GF fabrics.

**Figure 6 materials-16-05001-f006:**
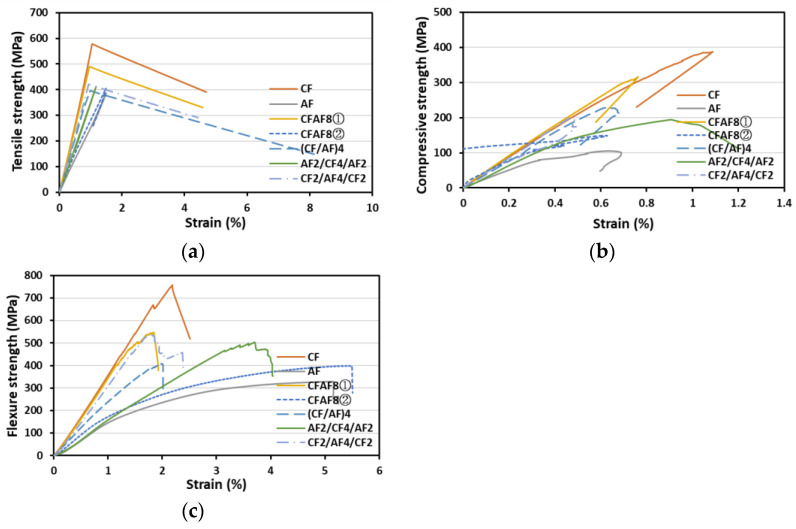
(**a**) T-S, (**b**) C-S, and (**c**) F-S graphs for the CF-AF fabrics.

**Figure 7 materials-16-05001-f007:**
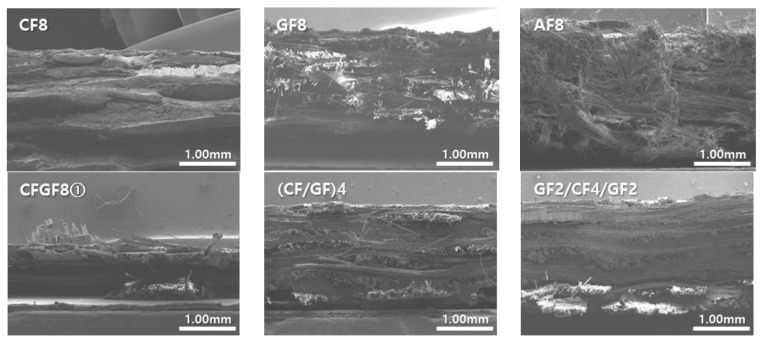
FRP fracture surfaces.

**Table 1 materials-16-05001-t001:** FRP stacking structure.

Sample	Name	Stacking Pattern	Mixing Ratio (*v/v*)
Reference fabric	CF8	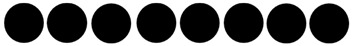	100	-
GF8, AF8	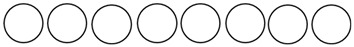	-	100
Hybrid fabric	CF2/GF4/CF2, CF2/AF4/CF2	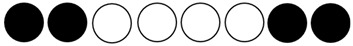	50	50
GF2/CF4/GF, CF2/GF4/CF2	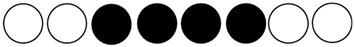	50	50
(CF/GF)4, (CF/AF)4	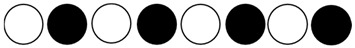	50	50
Blended fabric	CFGF8①,②, CFAF8①,②	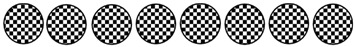	50	50

●: Carbon fiber fabric (CF), ○: Glass fiber fabric (GF) or aramid fiber fabric (AF), 
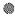
: Blended carbon–glass fiber fabric (CFGF) or blended carbon–aramid fiber fabric (CFAF).

**Table 2 materials-16-05001-t002:** FRP fiber/resin contents.

Category	Calculated Value				TGA		Thermal Expansion Coefficient
	Resin Content vol.%	Fiber Content vol.%	Resin Content wt.%	Fiber Content wt.%	Resin Content wt.%	Fiber Content wt.%	μm/m°C
Matrix resin							61.20
CF8	39.44	60.56	39.87	60.13	47.24	52.76	64.50
GF8	49.56	50.44	30.23	69.77	29.48	70.52	42.30
AF8	47.34	52.66	37.2	62.8	37.65	62.35	78.50
CFGF8①,②	42.84	57.16	27.99	72.01	36.81	63.19	54.80
(CF/GF)4	52.49	47.51	36.42	63.58	37.53	62.47	59.90
GF2/CF4/GF2	51.2	48.8	35.23	64.77	42.15	57.85	60.50
CF2/GF4/CF2	46.78	53.22	31.31	68.69	-	-	46.00
CFAF8①,②	52.99	47.01	42.05	57.95	-	-	84.80
(CF/AF)4	52.85	47.15	41.91	58.09	-	-	69.10
AF2/CF4/AF2	51.23	48.77	40.34	59.66	-	-	58.70
CF2/AF4/CF2	48.65	51.35	37.88	62.12	-	-	85.70

**Table 3 materials-16-05001-t003:** Tensile/compressive/flexural strength and strain of the reference fabrics.

Category	Tensile Strength (MPa)	Elasticity (GPa)	Strain (%)	Compressive Strength (MPa)	Elasticity (GPa)	Strain (%)	Flexural Strength (MPa)	Elasticity (GPa)	Strain (%)
CF8	568.17	53.88	0.97	381.04	48.32	0.87	658.23	33.58	1.91
GF8	251.56	24.06	1.31	224.52	26.6	0.9	339.7	17.63	2.27
AF8	379.9	25.76	1.61	102.72	21.54	0.88	325.76	13.01	5

**Table 4 materials-16-05001-t004:** Tensile, compressive, and flexural strengths for CF-GF mixing.

Category	Tensile Strength (MPa)	Tensile Modulus (GPa)	Strain (%)	Compressive Strength (MPa)	Tensile Modulus (GPa)	Strain (%)	Flexural Strength (MPa)	Tensile Modulus (GPa)	Strain (%)
CF8	568.17	33.58	0.97	381.04	48.32	0.87	658.23	53.88	1.91
CFGF8①	608.34	20.01	1.3	356.6	45.07	0.75	536.25	48.32	3.06
CFGF8②	428.31	17.3	1.71	286.88	37.62	0.81	584.2	29.06	3.99
(CF/GF)4	367.31	25.62	0.93	285.02	38.42	0.67	538.11	40.55	2.1
GF2/CF4/GF2	409.06	21.2	1.05	319.75	38	0.86	462.07	39.83	2.33
CF2/GF4/CF2	388.01	31.17	1.09	284.48	38.88	0.79	592.1	37.85	1.95

**Table 5 materials-16-05001-t005:** Tensile, compressive, and flexural strengths for CF-AF mixing.

Category	Tensile Strength (MPa)	Tensile Modulus (GPa)	Strain (%)	Compressive Strength (MPa)	Tensile Modulus (GPa)	Strain (%)	Flexural Strength (MPa)	Tensile Modulus (GPa)	Strain (%)
CF8	568.17	53.88	0.97	381.04	48.32	0.87	658.23	33.58	1.91
CFAF8①	516.3	54.1	0.97	300.95	42.55	0.72	542	32.21	1.82
CFAF8②	417.34	30.95	1.61	146.35	29.28	0.63	391.57	19.23	5.24
(CF/AF)4	386.36	44.35	0.94	216.27	36.11	0.67	428.65	24.8	2
AF2/CF4/AF2	412.85	37.06	1.13	235.88	35.42	0.73	519.13	15.1	3.48
CF2/AF4/CF2	399.14	46.05	0.77	233.99	40.95	0.61	545.5	29.33	1.84

**Table 6 materials-16-05001-t006:** Tensile strength, tensile modulus, and strain for hybrid and blended fabrics.

Authors	Stacking Method	Mixing	Tensile Strength	Tensile Modulus	Strain
			MPa	GPa	%
Sun et al. [13]	-	[C]_8_	504.73	-	-
	-	[B]_8_	413.50	-	-
	sandwich-stacking	[C_2_B_2_]_s_	354.39	-	1.07
	sandwich-stacking	[B_2_C_2_]_s_	385.22	-	1.19
	cross-stacking	[BCBC]_s_	437.15	-	1.26
Margabandu et al. [24]	-	[CC]_s_	301.17	22.84	1.45
	-	[JJ]_s_	54.98	6.47	1.24
	sandwich-stacking	[JC]_s_	234.68	15.35	2.74
	sandwich-stacking	[CJ]_s_	158.74	12.75	2.14
Chen et al. [25]	-	CC	679.1	59.2	1.5
	blended-stacking	C90G0	656.5	62.2	1.5
	blended-stacking	C0G90	483.7	22.1	2.7
	-	GG	478	22.1	2.7

C: Carbon fiber, B: Basalt fiber, J: Jute cottage, G: Glass fiber.

**Table 7 materials-16-05001-t007:** Blended FRP vertical deviation (є_y_).

Category	Unit	Gauge Length	CF8	CFGF8①	CFGF8②
Horizontal length change	mm	130	131.26	131.26	131.7
Vertical length change	mm	25	24.88	24.81	24.84
Vertical difference (ε_y_)	mm	-	0.12	0.19	0.16

## Data Availability

MDPI Research Data Policies.

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
