# Peer review of "Control of Mechanical Properties of FRP (Fiber-Reinforced Plastic) via Arrangement of High-Strength/High-Ductility Fiber in a Blended Fabric"

_materials, 2023, doi:10.3390/ma16145001_

Round 1
Reviewer 1 Report
The authors tried to improve the performance of FRP via different methods. The paper is generally good but it needs improvement. Followings should be carried out before acceptance:
The abstract should contain important results of the study.
Line 35. How strength of GFRP is low than steel?
Novelty is not clear. Add research significance
The importance of the use of FRP should be more emphasized using followings: influence of heat–cool cyclic exposure on the performance of fiber-reinforced high-strength concrete; buckling analysis of CNT-reinforced polymer composite beam using experimental and analytical methods; effects of stirrup spacing on shear performance of hybrid composite beams produced by pultruded GFRP profile infilled with reinforced concrete; the effects of eccentric web openings on the compressive performance of pultruded GFRP boxes wrapped with GFRP and CFRP sheets; compressive behavior of pultruded GFRP boxes with concentric openings strengthened by different composite wrappings
The reason fr selecting the stacking pattern should be explained.
Why 1:4 rati of resin and curing is utilized?
Compare your results with existing studies
Add photos for test setup? Did you use extronsometer?
Add photos of final product
Add some summary for conclucision
Add recent studies on this subject to introduction. There are many studies on the introduction for this topic.
Conclusion should be improved. The recommendation consdiering all test should be given for engineers.
Author Response
We response to the Reviewer's comments.(in file)

Author Response

(The authors gave the same response as above.)

Reviewer 3 Report
This paper shows the experimental results of various fiber-reinforced polymers using three types of fibers. Several lamination and woven conditions were adopted, and fundamental material tests followed ASTM standards. The results will be an important experimental result for the material design of fiber composite materials. This reviewer thinks the manuscript can be accepted with minor revisions.
1) The authors suggested "optimal arrangement" in the title. However, more suggestions were needed for the optimization. The authors must improve.
2) Usually, matrix resin was optimized to the fiber type to perform the fiber's performance well. Can this resin the authors adopted be used (recommended) for CF, GF, and AF? Moreover, the authors should explain how to choose the resin product.
3) In Figures 4 and 5, the stress-strain relations jumped (e.g., T-S slipped, C-S returned) after reaching the maximum stress with a large strain. What happened? And, are those behavior scientifically important?
4) This reviewer needs clarification that the tensile strength of CF8 is a bit lower than CFGF8-1. Theoretically, Pure CFRP should have the highest tensile strength. The authors should note the reason for these results. And this reviewer thinks the failure behavior must be checked carefully. Did not the stress concentration at the chucking position occur?
5) In the cases of hybrid fiber composites, the thermal expansion coefficient difference will relate to the thermal deformation, stress, etc. After curing, the authors should show the geometrical accuracy and comment on the thermal characteristics.
6) This reviewer thinks the literature review needs to be revised (improved).
Author Response
We answered the reviewer's comments.

Round 2
Reviewer 1 Report
The paper can be accepted
Author Response
Thank you for your response.
Reviewer 2 Report
Accept in present form.
Author Response
Thank you for your response.